# Sexual Dimorphism and Breed Characterization of Creole Hens through Biometric Canonical Discriminant Analysis across Ecuadorian Agroecological Areas

**DOI:** 10.3390/ani10010032

**Published:** 2019-12-22

**Authors:** Paula Alexandra Toalombo Vargas, Francisco Javier Navas González, Vincenzo Landi, José Manuel León Jurado, Juan Vicente Delgado Bermejo

**Affiliations:** 1Escuela Superior Politécnica de Chimborazo, Facultad de Ciencias Pecuarias, EC060155 Riobamba, Ecuador; paulasol37@yahoo.es; 2Department of Genetics, Faculty of Veterinary Sciences, University of Córdoba, 14071 Córdoba, Spain; juanviagr218@gmail.com; 3Animal Breeding Consulting, S.L., Córdoba Science and Technology Park Rabanales 21, 14071 Córdoba, Spain; 4Centro Agropecuario Provincial de Córdoba, Diputación Provincial de Córdoba, 14071 Córdoba, Spain; jomalejur@yahoo.es

**Keywords:** breed characterization, dual-purpose, agroecological regions, local products, ecuador, zoometry

## Abstract

**Simple Summary:**

The first step towards the protection and valorization of the genetic resources of a country is their definition. Although Ecuadorian zootechnical species are very diverse, they are scarcely characterized, and hence the efforts towards their protection are not as fruitful as they could be. The present paper approaches the biometric characterization of the Creole hen population in Ecuador through the study of sexual dimorphism and the differentiation of an agroecologically-based structured population using fourteen zoometric measures as differentiation criteria. Highlands region provinces of Cotopaxi and Tungurahua were the most zoometrically distant from the rest. However, Morona Santiago province population in the Amazonian region slightly differs from those in Guayas, Chimborazo and Bolívar in the Coastal and Highlands regions, respectively. The dual-purpose nature of Ecuadorian Creole hen resources enables the implementation of breeding programs that seek meeting a wider scope of public demands, through the definition of the agroecologically-based breed differentiated production of local hen eggs and meat.

**Abstract:**

Despite Ecuador having a wide biodiversity of zootechnical species, characterization studies of these genetic resources are scarce. The objective of this research was to perform the biometric characterization of the Creole hen population through 14 quantitative zoometric measures. We evaluated 207 hens and 37 roosters from Ecuador’s three agro-ecological regions: the Sierra (highlands) region (Bolivar, Chimborazo, Tungurahua and Cotopaxi provinces); the Costa (coastal) region (Guayas); and the Oriente Amazonian region (Morona Santiago). Sexual dimorphism was assessed using one-way analysis of variance (ANOVA). Body dimensions were generally significantly higher for males (*p* < 0.05), especially for length of head, beak, neck, dorsum, tarsus, thigh, leg, and middle finger. Then, individuals were biometrically clustered into populations after a stepwise canonical discriminant analysis (CDA) computing interpopulation Mahalanobis distances. Agroecologically-based structured populations were identified when zoometrical criteria were used to classify the animals. Cotopaxi and Tungurahua provinces were reported to be the most distant from the rest, with a slight differentiation of the Morona Santiago province population from those in Guayas, Chimborazo and Bolívar. Conclusively, Ecuadorian Creole hens were higher than longer contrasting light hen breeds, which favors their dual-purpose aptitude. Hence, the development of selection programs aimed at Ecuadorian differentiated entity of production of eggs and meat across agro-ecological areas is feasible.

## 1. Introduction

Genetic diversity of domestic hens existing across Ecuador is not only promoted by climatic stratification but also natural and human driven selection. From the red junglefowl (*Gallus gallus*), the most likely ancestor of these avian populations [1], the effects of natural selection may have resulted in a high heterogeneity and variability of the morphological characteristics of fowl, with a high potential to adapt to the different environmental conditions [2,3,4,5,6].

Contextually, Continental Ecuador comprises three agroecological regions, which are the Sierra (Andean highlands), the Amazonian Oriente (eastern rainforests), and the Costa (Pacific coastal lowlands), with five thermal or climatic floors: the warm floor (0–1000 m above mean sea level (mamsl) at 25 °C); the temperate floor (1000 to 2000 mamsl, at temperatures that range from 16 to 23 °C); the cold floor (2000 to 3000 mamsl at 12 °C); the paramo/moor floor (from 3000 to 4000 mamsl close to 0 °C); and the glacial floor (4000 mamsl < 0 °C).

Apart from naturally driven selection processes and natural migratory movements, genetic variability in local chicken populations may have been conducted as a result of human action. For instance, human-made migration processes [7] brought about the widespread distribution of poultry genetic material, given the size of animals was convenient and facilitated transport, favoring the expansion of these fowl across the different agroecological levels [3].

These factors led to genetic divergences contributing to poultry production under a family-run backyard system usually developed by each household’s women [3]. Husbandry practices characterized by the use of rustic animals in free range conditions with a low capital investment, which enables assuming a relatively low economic risk to implement an efficient productive management to produce high-biological-value protein sources such as meat and eggs [8,9]. Additionally, these products are preferred among consumers because of their pigmentation, taste, and lean quality of meat [10,11]; which translates into acceptable income that returns to each family, closing the cycle [12,13,14,15,16,17].

Breeds originating in the Old World were introduced to Latin American territories by the Spanish colonists and adapted to the different agroecological areas and conditions that they found, forming what has traditionally been addressed as Creole hen populations. For decades, these creole populations occupied local productive niches and evolved towards their current state, but still lacked the necessary characterization actions that may help consolidating and protecting them. In parallel, current breed development and formation until the XVII century provided the basic elements for the directed selection of our days and for the pursuit of concrete characteristics of interest to the farmer or producer. In this context, a new conglomerate of breeds and commercial lines formed in the first world were introduced into developing countries in an attempt to fulfil the growing market demands at a lower cost [18].

This global situation resulted in an alarming loss in the biodiversity of animals of zootechnical interest that the region faces nowadays. According to Food and Agriculture Organization (FAO), the endangerment risk that 81 percent of Latin America and the Caribbean avian breeds are exposed to is unknown [19], as even censuses are not appropriately registered. The increased risk of a population whose endangerment status is unknown is based on the fact that measures towards its protection are not implemented. In this regard, efforts are being made to maintain, conserve and, in turn, benefit from their most profitable or useful traits, such as disease or stress resistance, in commercial breeding plans [18].

Not only local hen breeds face a serious risk of extinction, but there is also a simultaneous loss of the traits that made them survive after the evolutionary process that they followed when they arrived in and adapted to the lands to which they were introduced. Creole hens present a good ability to scavenge and forage, have good maternal qualities, and are hardier than exotic breeds with higher survival rates and minimal care and attention requirements. This rusticity is one of those traits to positively influence avian zootechnical production, given its implication with the adaptation ability of animals to the environment in which they are produced.

After a period characterized by a lack of actions regarding local genetic resources conservation, with policies more likely focusing on intensive production, morphological characterization studies in fowl started being run again in Ecuador. These studies lay the basis for local resources conservation and breeding plans. Zoometric traits have widely been reported to depend on an inherited basis and to be suitable means of prediction for the live weight of the individuals [20,21,22]. Thus, they may play an important role in the subsequent performance of animal carcasses [23]; a relationship that translates to new potential selection criteria, seeking the maximization of the profitability of the products derived from such local genetic resources.

Despite the fact that research projects seeking the zoometrical characterization of Ecuadorian local hen breeds started being implemented using univariate analysis, there is still a patent lack of knowledge regarding the differentiation of such local populations, and hence policies towards the protection of such genetic resources cannot be implemented properly. Therefore, the aim of this study was to perform differentiated zoometric characterization of Creole hens through the application of a canonical discriminant analysis (CDA) to provide insights on the possible clustering patterns described by the population and into which subpopulations can be distinguished using Ecuadorian provinces as the criteria of origin [24]. Conclusively, this approach will enable quantification of the large existing phenotypic variability in the Ecuadorian creole hen population as a strategy to facilitate the rational development of such productively important avian local resources and their use, and to direct the implementation of conservation strategies aimed at ensuring their survival in the competitive world of poultry production and future consolidation as breeds.

## 2. Materials and Methods

### 2.1. Sample Size and Distribution

The whole sample comprised 281 fowl, 244 hens (84.84%) and 37 roosters (15.16%), evaluated across the three regions and six provinces, as follows: the Sierra Region (Andean highlands), with the provinces of Bolivar (31), Chimborazo (70), Tungurahua (35), Cotopaxi (32); the Amazonian Oriente Region (eastern rainforests) with Morona Santiago (38); and the Costa Region (Pacific coastal lowlands) with Guayas (28), respectively. Stevens [25] provided a very thorough discussion of the sample sizes that should be used to obtain reliable results for canonical discriminant analysis. Strong canonical correlations in the data (R > 0.7), even in cases of relatively small samples (around n = 50) will be detected most of the time. However, to obtain reliable estimates of the canonical factor loadings for interpretation, hence, to be able to draw valid conclusions, Stevens recommends that there should be at least 20 times as many cases as variables included in the analysis, if one wants to interpret the most significant canonical root only, as is the case in our study. To arrive at reliable estimates for two canonical roots, Barcikowski and Stevens [26] recommend, based on a Monte Carlo study, to include 40 to 60 times as many cases as variables.

### 2.2. Study Site Characterization and Sample Animals Management

The study was conducted under field conditions from January to December 2015 and from January 2017 to August 2018. The animals comprising the sample were raised and kept by backyard producers who did not present evidences of crosses with commercial lines among the effectives of their farms, following a randomized design. The map of provinces and climatic floors of Ecuador are shown in Figure 1 and Figure 2. Agroecological zones are detailed in Table 1.

Animals were reared under extensive backyard conditions, and were not vaccinated against viruses or parasites such as coccidia, nor treated against parasites. Chickens were fed on organic corn and were occasionally supplemented with household wastes, vegetables, and other sources of minerals from each area. Antibiotics and multivitamins were not administered.

### 2.3. Biometric Data Collection

Biometrical analysis was performed on each animal, measuring the sixteen quantitative variables proposed by FAO [28]. A summary of the biometric variables measured and the procedure followed is shown in Table 2. Quantitative data was obtained using a digital scale, a gauge with 0.02 mm accuracy, and a tape measure. All the biometric information was collected in a structured file including georeferencing for each producer together with zoometric measurements.

### 2.4. Statistical Analysis

#### 2.4.1. Descriptive Statistics, ANOVA and Waller-Duncan Post-Hoc Test

Mean and standard deviation of each measurement were computed for each sex and province. one-way analysis of variance (ANOVA) was carried out using the MEANS Statement from the PROC GLM routine of the S.A.S. 9.4 software [31] to determine the existence of differences in the means for the fourteen variables measured between males and females and across provinces. Then, the WALLER option was used to perform post hoc Waller-Duncan k-ratio t test on all main effect to measure specific differences between pairs of means (*p* < 0.05). Waller and Duncan [32] and Duncan [33] take an approach to multiple comparisons that differs from all the methods previously discussed in minimizing the Bayes risk under additive loss rather than controlling type I error rates. Furthermore, this range test uses the harmonic mean of the sample size, which makes it preferable if the sample sizes are unequal [34].

#### 2.4.2. Canonical Discriminant Analysis (CDA)

The multivariate technique involved the use of canonical discriminant analysis on the 14 biometric measurements using the province to which each animal belonged as a labeling classification criteria, to identify the variation provided by the different variables measured under study, and to establish clusters that may identify and outline subpopulations [35,36,37,38]. Hence, we determined the percentage of correctly allocated individuals in their populations of origin in comparison to those animals which were statistically misclassified or attributed to a different province from the one in which they were sampled, to discover a linear combination of quantitative morphological variables that provide maximum separation between the potentially existing different populations when the classification criterion was the province in which the animals were located. CDA was also used to plot pairs of canonical variables to help visually interpret group differences. Variable selection was performed using Forward Stepwise (FSTEP) multinomial logistic regression algorithms. Canonical discriminant analysis was performed using the CANDISC Procedure from the PROC CANDISC routine of the S.A.S. 9.4 software [31].

##### Canonical Correlation Dimension Determination

Canonical correlation is a form of correlation relating two sets of variables. The maximum number of canonical correlations between two sets of variables is the number of variables in the smaller set. The first canonical correlation is always the one which explains most of the relationship [39]. The canonical correlations are interpreted as Pearson’s r, and hence their square is the percentage of variance in one set of variables explained by the other set along the dimension represented by the given canonical correlation (usually the first); that is, Rc-squared is the percentage of shared variance along this dimension [40]. As a rule of thumb, some researchers state that a dimension will be of interest if its canonical correlation is 0.30 or higher, corresponding to about 10% of variance explained. Despite this, some researchers report just the first canonical correlation; it is recommended that all meaningful and interpretable canonical correlations are reported [41].

##### Canonical Discriminant Analysis Efficiency

Wilks’ Lambda test assesses which variables significantly contribute to the discriminant function. As a rule of thumb, the closer Wilks’ lambda is to 0, the more the variable contributes to the discriminant function. The significance of Wilk’s Lambda can be tested using Chi-Square, and then, if the *p*-value if less than 0.05, we can conclude that the corresponding function explains group adscription well [42]. For small sample sizes or a small number of treatments, the limiting chi-squared or normal distributions may not adequately describe the actual probability distributions of the test statistics. Here, a finite approximation may be more appropriate than using the limiting distribution. One such method is Fisher’s F approximation for Wilks’ lambda by Rao [43] as developed in Chávez [44]. According to these authors, under normality conditions, this procedure performs more accurately than a χ^2^ approximation [45].

##### Canonical Discriminant Analysis Model Reliability

Box’s M is used to test the assumption of equal covariance matrices in Multivariate Analysis of Variance (MANOVA) and discriminant function analysis (DFA). Box’s M has very little power [46] for small sample sizes; hence when we work with a small sample a nonsignificant result may not necessarily indicate that the covariance matrices are equal. In contrast, for large samples a statistically significant result can be reported when it does not actually exist. To address this particular issue, a smaller alpha level (*p* < 0.001) is recommended [47]. Some authors suggest that Box’s M is highly sensitive, hence unless *p* < 0.001 and sample sizes are unequal, we should ignore it. However, if the results are significant and you have unequal sample sizes, the test is not robust [48].

In multiple regression, another assumption that should be tested for is multicollinearity. The variance inflation factor (VIF) is used as an indicator of multicollinearity. Computationally, it is defined as the reciprocal of tolerance: 1/(1 − R^2^).

Various recommendations for acceptable levels of VIF have been published in the literature. Perhaps most commonly, a value of 10 has been recommended as the maximum level of VIF [49,50,51,52]. The VIF recommendation of 10 corresponds to the tolerance recommendation of 0.10 (1/0.10 = 10). However, a recommended maximum VIF value of 5 [53] and even 4 [54] can be found in the literature.

For example, a VIF of 8 implies that the standard errors are larger by a factor of 8 than would otherwise be the case, if there were no inter-correlations between the predictor of interest and the remaining predictor variables included in the multiple regression analysis.

##### Canonical Coefficients and Loading Interpretation and Spatial Representation

A preliminary principal component analysis (PCA) was performed to minimize overall variables into few meaningful variables that contributed most to variations in the populations. As a result, half wing radius ulna length (hwrul) and distal phalanx wing length (dpwl) were discarded, given they reported a component loading lower than |0.5| which suggested their redundant confounding nature, which may base on the fact that they comprise the total length of the wing, defined by proximal humerus wing length (phwl). Discriminant function analysis was used to determine percentage assignment of individuals into their own populations.

The traditional approach to interpreting discriminant functions examines the sign and magnitude of the standardized discriminant weight (also referred to as a discriminant coefficient) assigned to each variable in computing the discriminant functions. Small weights may indicate either that a certain variable is irrelevant in determining a relationship or that it has been discarded because of a high degree of multicollinearity.

Discriminant loadings reflect the variance that the independent variables share with the discriminant function. In this regard, they can be interpreted like factor loadings in assessing the relative contribution of each independent variable to the discriminant function.

In either simultaneous or stepwise discriminant analysis, variables that exhibit a loading of >|0.40| or higher are considered substantive, indicating substantive discriminating variables. With stepwise procedures, this determination is supplemented because the technique prevents nonsignificant variables from entering the function. However, multicollinearity and other factors may preclude a variable from entering the equation, which does not necessarily mean that it does not have a substantial effect. Loadings are considered to have relatively higher validity than weights as a means of interpreting the discriminating power of independent variables because of their correlational nature.

Standardized coefficients allow you to compare variables measured on different scales. Coefficients with large absolute values correspond to variables with greater discriminating ability. Also, discriminant scores can be computed by using the standardized discriminant function coefficients applied to data that have been centered and divided by the pooled within-cell standard deviations for the predictor variables, as discussed in IBM Corp. [55].

The data were standardized following standard procedures of Manly [56] before squared Mahalanobis distances and principal component analysis were computed. Squared Mahalanobis distances were computed between populations using the following formula:
(1)$$D_{ij}^{2}=\left(\bar{Y_{i}}-\bar{Y_{j}}\right)\;COV^{-1}\left(\bar{Y_{i}}-\bar{Y_{j}}\right)$$
where $D_{ij}^{2}$ is the distance between population i and j, COV^−1^ is the inverse of the covariance matrix of measured variable x, and $\bar{Y_{i}}$ and $\bar{Y_{j}}$ are the means of variable x in the ith and jth populations, respectively. The squared Mahalanobis distance matrix was converted into a Euclidean distances matrix and used to build a dendrogram using unweighted pair-group method using arithmetic mean (UPGMA) via agglomerative hierarchical cluster procedure with the software DendroUPGMA by Garcia-Vallvé and Pere Puigbo [57]. The Mahalanobis squared distance, defined as the square of the distance between the measures of the standardized values of Z (centroids), was used this way to verify whether there were significant differences between provinces [58].

##### Discriminant Function Cross-Validation

To establish whether the percentage of correctly classified cases is high enough to consider that the discriminant functions issue valid results, as a form of significance we can use leave-one-out cross-validation option. Classification accuracy achieved by discriminant analysis should be at least 25% greater than that obtained by chance.

These results can be supported by Press’ Q statistic. This parameter can be used to compare the discriminating power of our function to a model classifying individuals at random (50% of the cases correctly classified), as follows
Press Q = [N − (nK)]^2^/N(K − 1) = 227 − (221*6)^2^/227(6 − 1) = 1064.1418(2)
where N is the number of individuals in the sample, n is the number of observations correctly classified (as a coefficient ranging from 0 to 1), and k is the number of groups.

The next step is to compute the critical value, which equals the chi-square value at 1 degree of freedom. It is advisable to let alpha equal 0.05. When Q exceeds this critical value, classification can be regarded as significantly better than chance, thereby supporting cross-validation.

## 3. Results

### 3.1. Descriptive Statistics, ANOVA and Waller-Duncan Post-Hoc Test

Morphometric analysis indicated highly significantly differences when males were compared to females, as shown in Table 3.

Table 4 shows a summary of the significant results of ANOVA and post hoc Waller-Duncan test for the zoometric characteristics of Ecuadorian creole hens across the provinces. The variables analyzed have a very high and variable coefficient of variation across biometric traits.

### 3.2. Canonical Discriminant Analysis

#### 3.2.1. Canonical Discriminant Analysis Model Reliability

The value of *p* < 0.05 obtained for Box’s M test means the data did not differ significantly from multivariate normal and we could proceed with the analysis. Wilk’s lambda statistic was used to assess whether canonical discriminating functions contributed significantly to the separation of treatments, that is, it was used to test the meaning of the discriminating function Table 5.

All zoometric variables were included at a preliminary stage of the analysis performed in this study. Tolerance (1/R^2^) and variance inflation factor (VIF) were analyzed to identify those variables that were responsible for multicollinearity between variables. This analysis revealed that the variables proximal humerus wing length (phwl), half wing radius ulna length (hwrul) and distal phalanx wing length (dpwl), turned out to be highly related (VIF > 4). Therefore, we decided to retain distal phalanx wing length (dpwl) in the analysis, because that measure results from the combination of proximal humerus wing length (phwl) and half wing radius ulna length (hwrul), given their lower VIF. After the removal of redundant variables, the results for tolerance and VIF can be seen in Table 6.

#### 3.2.2. Canonical Coefficients and Loading Interpretation and Spatial Representation

The canonical discriminant analysis identified five discriminating canonical functions. The first had a high discriminatory power, as denoted by the eigenvalue of 5.422. The results are presented in Table 7. The first function obtained explains 91.9% of total variance. The fifth function contributes to the explanation of variance with 49.9% of the information to the analysis, that is, relatively low.

The results for the tests of equality of group means to test for differences across provinces once redundant variables have been removed are shown in Table 8. The greater the value of F and the lower the value for Wilks’ Lambda, the better the discriminating power a certain variable has and the lower the rank position it presents. Those variables presenting equal values of lambda and F had equivalent discriminatory power, as shown by beak and dorsal length. When this happens, it is necessary check whether these similarities are based on a multicollinearity problem or are because the variables, indeed, have a similar discriminant power.

Once F and Wilks’ Lambda had been assessed, we evaluated the magnitude of standardized and non-standardized coefficients, reported in Table 9, to determine whether there had been a reduction in the discriminant power of individual variables as a result of multicollinearity between pairs, which implies a reduction in the separate discriminant power of each of the two variables involved in the multicollinear relationship.

As shown in Table 8 for the variables neck and dorsal length, we observe that standardized coefficients fell below 0.4, hence, hence there was a decrease in the discriminating power of the non-individual function as a result of the effect of multicollinearity, that is, neck and dorsal length variables being related and explaining a somehow redundant fraction of variability.

The greater the reduction in the standardized coefficient, the more important the multicollinearity problem between variables holding similar Wilks’ lambda and F values. Standardized coefficients are shown in Table 9. According to Hair Jr [59], absolute values below |0.3| are indicative of multicollinearity problems when F and Wilks’ Lambda have been previously checked to be similar for a certain pair of variables.

Unstandardized coefficients, calculated on raw scores for each variable, are of most use when the investigator seeks to cross-validate or replicate the results of a discriminant analysis or to assign previously unclassified subjects or elements to a group. As we are assessing the potential misclassification of individuals belonging to previously defined populations as a way to define such populations themselves, we must interpret standardized coefficients, and hence unstandardized coefficients were discarded [60]. Furthermore, the unstandardized coefficients cannot be used to compare variables or to determine what variables play the greatest role in group discrimination because the scaling for each of the discriminator variables (i.e., their means and standard deviations) usually differ. The maximum number of canonical discriminant functions generated is equal to the number of groups minus one. In the present study, the number of canonical discriminant functions was 5 for each series, as we used the six provinces as a labelling criterion. After the evaluation of standardized coefficients, the resulting discriminant functions were as follows:
F1: (−1.0371) × mfpl + (1.2468) × metl + (0.5107) × legcF2: (−0.6800) × legl + (−0.6413) × thil + (0.4506) × dpwl + (0.4387) × venlF3: (−0.4368) × dpwl + (0.6318) × chpl + (−0.8659) × venl + (0.4596) × picl + (0.4228) × crewF4: (−0.6428) × legc + (0.7568) × healF5: (0.4576) × thil + (0.5706) × peal + (0.5273) × necl + (−0.4764) × crel

To determine which is the variable that we have to discard out of each pair for which a multicollinearity problem has been detected, we checked discriminant loadings, which are presented in Table 10. Discriminant loadings measure the existing linear correlation between each independent variable and the discriminant function, reflecting the variance that the independent variables share with the discriminant function. In this regard, they can be interpreted like factor loadings in assessing the relative contribution of each independent variable to the discriminant function. A graphical representation of discriminant loadings is shown in Figure 3, with those variables whose vector extends further apart from the origin being the most representative discriminating ones.

A territorial map was created by plotting the discriminating values for each observation (Z) for the first function on the x axis and those values for the second discriminant function on the y axis. Figure 4 graphically depicts the canonical discriminant analysis of individuals across the six sampling provinces. Cotopaxi and Tungurahua provinces are evidently independent, while the populations from the provinces of Guayas, Chimborazo and Bolívar displayed significant overlap. Further away from the latter three, we find the population from the province of Morona Santiago.

Centroids designed the central observation for each province group. The probability that an unknown case belongs to a particular group was calculated by measuring the relative distance of Mahalanobis to the centroid of a population. To compute discriminant scores or centroids, we substituted the mean for each possible province in the three first dimensions [61]. Then, to calculate the optimal cut-off point, that is, the probability of classification we followed the procedures in Hair, Black, Babin and Anderson [52]. Then we could determine whether a certain case was appropriately classified.

It can be observed that the provinces of Cotopaxi and Tungurahua are located in different places on the Cartesian plane, that is, remote from each other. The opposite situation happened with the provinces of Bolívar, Guayas and Chimborazo, and it can also be observed that Morona Santiago is slightly separated from the three latter provinces.

#### 3.2.3. Discriminant Function Cross-Validation

When classification and leave-one-out cross-validation matrices are evaluated, it can be observed that 97.14% has been estimated to be correctly classified for Bolivar, with 88.57% being validated for the same province. For Chimborazo, 93.33% has been correctly classified, with 89.33% validated. For Cotopaxi, Guayas, Morona Santiago and Tungurahua the total of observation was appropriately classified with 90.63%, 50%, 80% and 94.44% being validated, respectively.

Cross-validation reported a result for Press Q parameter of 1064.1418 (Press Q = [N − (nK)]^2^/N(K − 1) = 227 − (221 × 6)^2^/227(6 − 1)). Hence, Q was above 6.63 (significance level of 0.01), Chi^2′^s critical value for a degree of freedom at a chosen confidence level. Predictions were significantly better than chance, according which it would have a correct classification rate of 50%

The absolute values of Mahalanobis’ distances between the local populations of the six provinces involved in the analysis are shown in Table 11. The shortest distance is found between Bolivar and Chimborazo, while the longest distances are those found between the Province of Cotopaxi and the rest. Contrastingly, the distances of Tungurahua with Bolívar, Chimborazo, Cotopaxi are similar. The level of statistical significance for all Mahalanobis distances was high and similar (*p* < 0.0001).

Mahalanobis’ distance analysis is based on the analysis of generalized squared Euclidean distances adjusted for unequal variances. The Mahalanobis distance (D^2^), defined as the square of the distance between the measures of the standardized values of Z, was used to verify whether there were significant differences between the provinces. Thus, the greater the value of the distance, the greater the distance between the means of the provinces considered as well [58]. As can be seen in the dendrogram Figure 5, the provinces of Bolivar-Chimborazo, Guayas-Morona Santiago, and Cotopaxi-Tungurahua are populations represented as subpopulations.

The shortest Euclidean distance was observed between the provinces of Bolivar and Chimborazo; whereas the opposite happened between the Province of Cotopaxi and the others. The distances between Tungurahua and Bolívar, Chimborazo, or Cotopaxi are similar. In contrast, Morona Santiago is slightly far from the provinces of Guayas, Chimborazo, Bolívar.

## 4. Discussion

The morphometric measurements show highly significant differences in relation to sex, as reported by Yakubu and Salako [62] in indigenous chickens in Nigeria, which reported such differences to be based on the hormonal effects of sex that condition growth. These results were consistent with similar results in the literature [6,63].

The high coefficient of variation observed in the results is similar to that reported in different populations of chickens in Mexico [64], and also in indigenous chickens in Nigeria [61], which demonstrates the variability of the morphometry in the birds studied, which may be due to genetic divergence processes followed by the populations, such as migration [65], which resulted in the morphological modification of the populations to adapt to the characteristics of the different environments and the orography to which the birds were introduced [63,66,67,68].

Measurements for head length are higher than those measured in Batsi Alak Hens of Mexico whose mean for males and females varied from 4.16 to 4.6 cm [67], and lower than those reported for Yoruba ecotypes of Nigeria with an average of 9.90 cm [68].

In terms of crest length, the values are lower than those measured in indigenous Nigerian roosters and higher than those indicated for hens of the same country [62]. Likewise, Yoruba and Fulani ecotypes [67] reported a similar value to that reported for the province of Cotopaxi when comparing fowl in general, without considering males and females separately [69]. However, in autochthonous Catalonian chicken breeds, such as Patridged Penedesenca and Blonde Empordanesa, we observe crest sizes double those measured in the present study [70].

For beak length, the values are analogous to those found in Botswana hens. This could be supported by the fact that both studies were conducted across three agroecological regions [71]. However, higher values were reported by Yakubu and Salako [62] in indigenous Nigerian fowl for males and females, as well as in native Catalonian breeds Partridged Penedesenca and Blonde Empordanesa. The opposite situation was described by Batsi Alak hens from Mexico [67] and Fulani ecotypes of Nigeria [68], which reported similar mean values to those of Bolivar, Guayas, Morona Santiago. This could be ascribed to the similarity between the climates of the locations in which the study took place.

For the neck length trait, the values found were equivalent to those found in Partridged Penedesenca and Blonde Empordanesa [70] and Yoruba and Fulani Nigerian ecotypes [68]; while lower values were found in indigenous Nigerian hens for both males and females [62]. The highest values in literature were reported for Batsi Alak Hens from Mexico with measures of 19 to 17 cm in males and females, respectively [67].

Body and dorsal lengths along with head length have been related in literature to the potential of animals for egg production [68]. When these data are compared with those reported by Moazami-Goudarzi [5] and their studies of local Tanzanian chicken ecotypes, the values for males of the Singamagazi ecotype were slightly higher than the average reported for Morona Santiago hens, but comparable to the males of the Kuchi ecotype. However, for the male of Mbeya ecotype and female of the Singamagazi ecotypes [5], these values were similar to those reported for the Patridged Penedesenca hens and for the Blonde Empordanesa [70] and equivalent to those found in Bolívar, Chimborazo, and Guayas. Studies conducted on Botswanan hens across three different agroecological areas found similar values for females and males [71] to those measured in Cotopaxi and Tungurahua. In turn, Yakubu and Salako [62] reported higher average measurements than those of Ecuadorian creole hens found in this study or those found for the Fulani and Yoruba.

Higher values for the ventral length variable are found in native breeds of Partridged Penedesenca, Blonde Empordanesa hens [5], and Yarubi and Fulani ecotypes [68]. The thoracic perimeter variable is a good indicator of meat yield in most species of poultry [68]. Higher values were found than those obtained for Nigerian males and females [62]. However, native Nigerian birds bred for research purposes belonging to the Anak Titan ecotype [72] and the Yoruba and Fulani ecotypes [68] reported similar values to those found for Chimborazo individuals.

The Batsi Alak hens in Mexico reached values similar to those reported in Bolívar, Chimborazo and Cotopaxi [67]. These values were common in backyard hens in Mexico [63]. The measurements for half wing radius ulna length (hwrul) were shorter than those for the males and females of the Batsi Alak hen breed from Mexico [67]. However, the values for distal phalanx wing length (dpwl), were similar to those observed in Batsi Alak hens from Mexico for both males and females. This may be due to the fact that the study was carried out at an altitude of 1200 to 2760 m above mean sea level, similar conditions found at the location where our study took place [67]. Thigh length in Partridged Penedesenca and Blonde Empordanesa hens [70] were similar to those from Ecuadorian fowl from Cotopaxi and Tungurahua provinces, but shorter than the Nigerian Yoruba ecotype hens [68].

Regarding the circumference of the leg, the indigenous hens of Nigeria reported similar values to those measured in Bolívar and Chimborazo and slightly similar ones to those reported for the hens of Cotopaxi. The dimensions of the leg have been related in literature with the type of production, with those animals presenting higher dimensions (both in width and length), being more appropriate to suit the requirements for meat production and characteristic of carnic breeds [68]. Tanzanian local chicken ecotypes presented higher tarso-metatarsal lengths [5] for Singamagazi and Kuchi than those of Ecuadorian Creole hens. However, females of the same ecotypes reported similar average values [73] to those measured in Morona Santiago hens.

Similarly, males of the Ching’wekwe ecotype and females of the same ecotype and female of the Morogo ecotype presented similar values to those of the provinces of Guayas and Tungurahua, but much higher than those of the province of Cotopaxi. Likewise, this value was similar to that reported by Nigerian birds [62], which may support the fact that ecotype, may be strongly conditioned by the agroecological conditions in the area in which these avian population are based.

Similarly, larger sizes were recorded for breeds such as the Partridged Penedesenca and Blonde Empordanesa with an average of 8 cm, which may be because the birds studied also come from four different eco-climates, homologous to those of the areas considered in this research.

For birds from India [74], average values were similar to those measured in Bolívar and Chimborazo. Long tarsi have been associated with dry regions and flat topographies, as they allow birds to travel long distances in search for food, unlike birds with short tarsi, which could be attributed to the effects of natural selection [72].

Short tarsi have been identified with a greater ability to escape from predators [74], hence, they have been directly related to processes of adaptation and improvement of survival. Functionally, from a productive point of view, tall animals tend to be destined for meat production and small animals for egg production [68]. In addition, the length of tarsi may be related to the prediction of live weight in the field, as reported by some authors [75,76,77]. Ecuadorian creole hens present morphological traits which would made them more prone to produce eggs. However, the dimensions of certain morphological variables could make them suitable for meat production, so it is essential to implement breeding programs to select and direct crosses that allow ecotypes to be obtained by classifying individuals depending on to their productive potential.

Canonical discriminant analysis suggests that the variables perimeter of the leg (legc), metatarsus tarsus length (metl) and middle finger phalanx length (mfpl) were the ones that had the greatest discriminatory capacity between provinces. The results revealed the presence of wide ranges of variation within and among Creole hens in Ecuador. However, four large population blocs could be identified, namely Cotopaxi, Tungurahua, Morona Santiago, and that comprising the populations of Bolivar, Chimborazo and Guayas, a fact that could be attributed to the different conditions found across the various agroecological zones in the country, ethnic groups handling these resources and cultural implications that they have, along with the huge migration events suffered by these resources when facing natural and/or man-made challenges.

Toalombo, et al. [78] identified a common pattern of haplogroups of mitochondrial DNA for Ecuadorian chickens reared across Ecuadorian agroecological systems, which suggests these animals may belong to the same maternal lineages. In fact, the maternal origin for these populations could presumably be attributed to pre-Columbian Asiatic matrilines or Iberian matrilines arriving during the Spanish colonization. Furthermore, mitochondrial findings support the fact that current Ecuadorian local chickens do not show maternal influences from commercial lines and maintain high levels of genetic diversity without evidence of genetic drift and/or population bottlenecks. This high diversity may be owing to internal heterogeneity, which, as suggested by the results in the present study, may be promoted by the breeding policies that are carried out. Additionally, the patterns found by canonical discriminant analysis are supported by the results reported by Toalombo et al. [78], as a certain internal substructure can be found. However, the absence of any breeding program, registers, or zootechnic management produces a high fragmentation of the potential Ecuadorian breeds.

The most likely reason for which Morona Santiago, Bolívar, Chimborazo, and Guayas populations present a mixed structure which does not permit their complete segregation from one another may have its basis on the fact that poultry farmers in this province are prone to preserve their birds, avoiding the introduction of individuals from external populations, despite the fact that poultry production has already been developed in their areas.

The provinces of Morona Santiago, Bolívar, and Chimborazo are nearby to each other, sin contrast to the province of Guayas, located in the Costa Region. However, there is a provincial road that joins Chimborazo (Sierra) and Guayas (Costa). It should be noted that agricultural fairs are likely to take place along the road connecting both provinces. These events act as exchange centers of genetic material, which is mainly performed with minor species such as birds, given the considerable ease to transport such resources. The same would happen with Morona Santiago, which is located in the Ecuadorian Amazon, but near Chimborazo, which is geographically located in the center of the country. The provinces in question are characterized by their agricultural and livestock background, such that 13% of the population is engaged in poultry production activity; hence, it could be assumed that the populations maintain the genetics of their fowl over time.

The connection with Guayas, as stated by the General Secretariat of the Andean Community in 2009, lays on the fact that during the seventies, there was a reduction in domestic agricultural-livestock production, which led to the migration of the inhabitants of Nabuzo-Penipe (Chimborazo) to the coast of Ecuador. This social movement implied people carried easily-used animal species such as hens along with them. Complementarily, as a result of the eruptions of the Tungurahua volcano, constant since the late 1990s [79], the Canton Penipe experienced immigration from the surrounding populations, which was also noticeable in Nabuzo (Chimborazo), the area least affected by volcanic ash. This confirms that the genetic material did not suffer such migratory pressure and hence its resources remain intact, which in turn explains the clustering revealed by the canonical discriminant analysis and which may outline the same genetic structure.

## 5. Conclusions

The results revealed the presence of wide ranges of variation within and among Creole hens in Ecuador. However, four large population blocs could be identified, namely Cotopaxi, Tungurahua, Morona Santiago, and that comprising the populations of Bolivar, Chimborazo, and Guayas, a fact that could be attributed to the different conditions found across the various agroecological zones in the country, the ethnic groups handling these resources and cultural implications that they have, along with the huge migration events suffered by these resources when facing natural and/or man-made challenges. This addresses the high number of opportunities to implement programs for the promotion, conservation, and genetic improvement of these local resources, through their selection and crossing after their definition and characterization, since their inherent resistance and adaptation to the different environmental conditions allow the definition of technical and scientific strategies to exploit their productive potential. In addition, the local nature of these resources should be highlighted as intangible ancestral value in the field of food sovereignty, and should be inserted within the productive policies at a governmental level, as has already been done with other species, with the aim of improving rural livelihoods and meeting the growing demand for poultry products.

## Figures and Tables

**Figure 1 animals-10-00032-f001:**
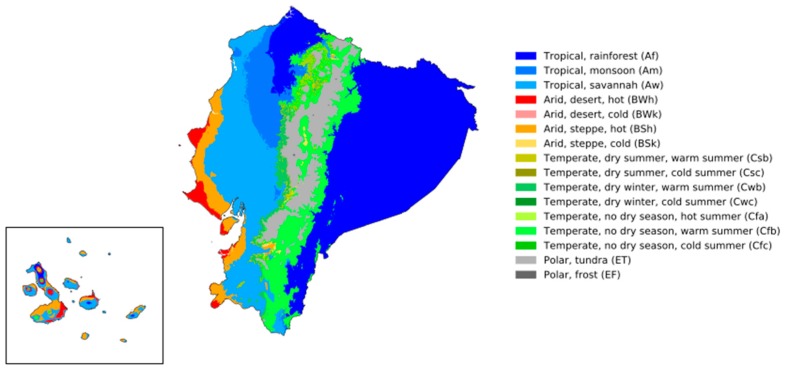
Köppen-Geiger climate classification map for Ecuador (1980–2016), accessed from Beck et al. [27].

**Figure 2 animals-10-00032-f002:**
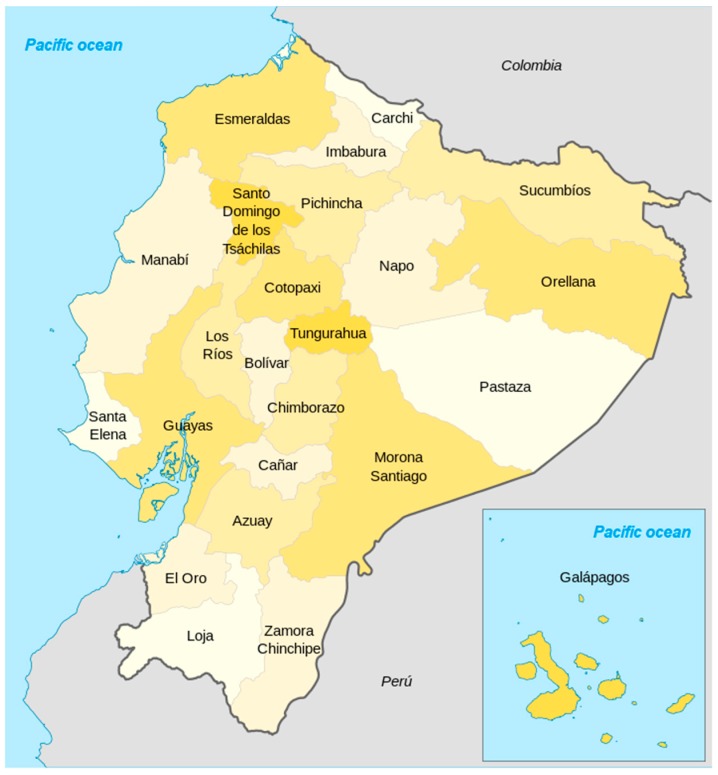
Administrative division of Ecuador in differents shades of yellow, accessed from Manuel Balarezo.

**Figure 3 animals-10-00032-f003:**
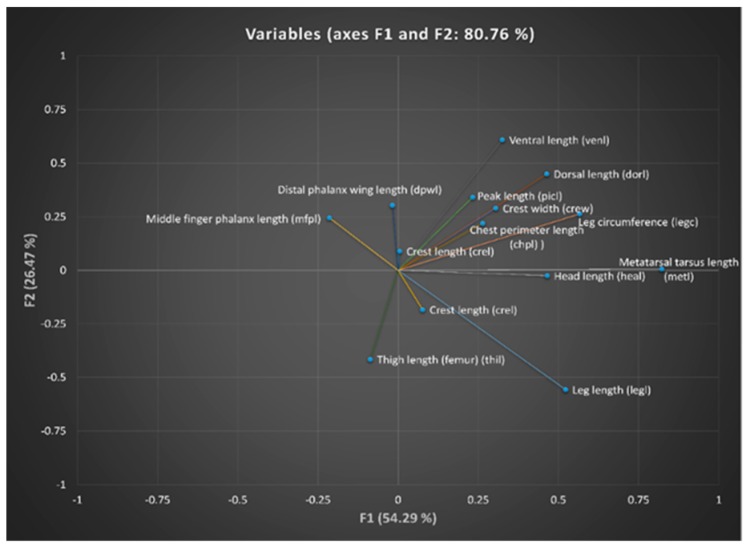
Vector plot of discriminant loadings for biometric variables.

**Figure 4 animals-10-00032-f004:**
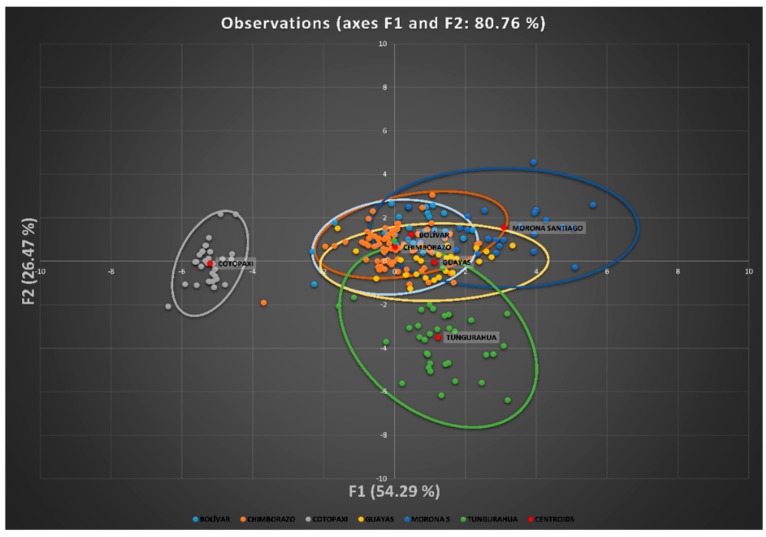
Territorial map depicting the results of the canonical discriminant analysis of individuals across the six sampling provinces.

**Figure 5 animals-10-00032-f005:**
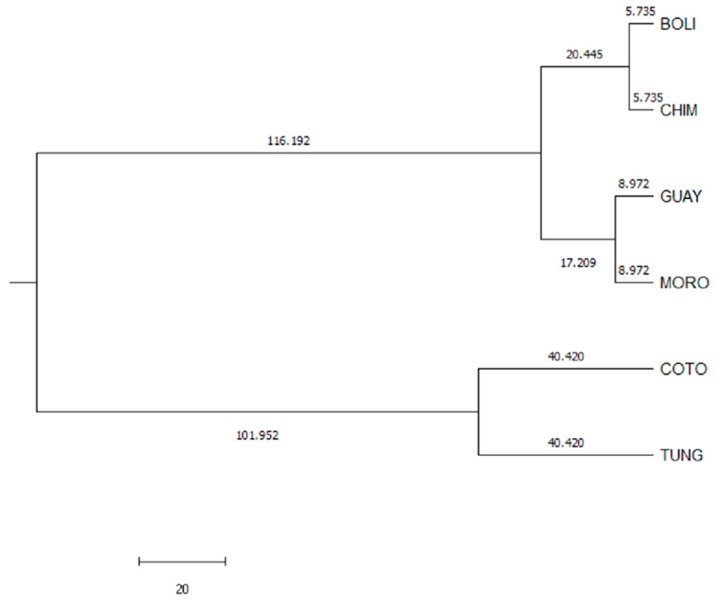
Euclidean distances dendrogram for the creole hen populations across provinces (BOLI: Bolívar, CHIM: Chimborazo, GUAY: Guayas, MORO: Morona Santiago, COTO: Cotopaxi and TUNG: Tungurahua).

**Table 1 animals-10-00032-t001:** Sampled provinces (climate and altitude) considered for the characterization of the Ecuadorian Creole hen.

Province	Climate	Altitude Measured in Meters above Mean Sea Level (MAMSL)
Bolívar	Tropical Megathermal Semi-Wet, Tropical	327
	Subtropical	1469
	Semi-wet to humid equatorial mesothermal	1500
Chimborazo	Semi-wet mesothermal	2700
	Temperate	2799
	Temperate	2800
	High mountain cold equatorial	2979
	High and upper montane	3341
	Equatorial cold semi-wet high mountain and high mountain cold equatorial	4200
	Equatorial cold semi-wet high mountain and high mountain cold equatorial	4300
Guayas	Warm and rainy	700
	Subtropical	1623
Tungurahua	Tropical rainy	1820
	Sub-Andean tempering, Andean and glacial cold	2500
	Temperate	2600
	Warm and dry	3400
Cotopaxi	Tropical rainy	1900
	Temperate and cold	2500
	Temperate	2938
	Wet tempering	2806
	Temperate	2917
	Temperate	2971
Morona Santiago	Warm humid	1030
	Warm humid	1199
	Tropical	1300
	Warm humid	2347

**Table 2 animals-10-00032-t002:** Biometric variables proposed by FAO and measuring procedure to obtain them from the animals.

Variable	How to Measure It
Head length (heal)	Taken between the most protruding point of the occipital and the frontal (lacrimal) bone.
Crest length (crel)	Taken following the direction of the skull.
Crest width (crew) ^a^	From the base of the head until the crest ends at the top of the face, following the opposite direction of the skull.
Beak length (peal)	In a caudo-cranial direction, from the base of the beak to the tip of the beak.
Neck length (necl)	Distance from the base of the neck to the chest.
Dorsal length (dorl)	Between the first thoracic vertebra to the region of the pygostyle (tail).
Ventral length (venl)	Length of the sternal region (keel).
Chest perimeter length (chpl)	Taken from the most declining part of the base of the cross, passing through the ventral base of the sternum and returning to the base of the cross, forming a straight circle around the coastal plains.
Proximal humerus wing length (phwl)	From the binding of the humerus with the spine to the termination of the humerus.
Half wing radius ulna length (hwrul)	From the union of the humerus with the radius and the ulna to the termination of them.
Distal phalanx wing length (dpwl)	Understood from the union of the radius and the ulna with the phalanges until the completion of them.
Thigh length (femur) (thil)	Distance from the middle region of the coxal bone to the knee joint.
Leg length (legl)	Distance between knee and tarsus joints.
Leg circumference (legc)	Measurement taken in the most prominent part of the leg
Metatarsal tarsus length (metl)	Distance between calcaneus and ankle
Middle finger phalanx length (mfpl)	Distance between the tarsus joint and the origin of the fourth finger.

^a^ Modified by the premises in Lázaro Galicia, et al. [29] and Estrada Mora, et al. [30].

**Table 3 animals-10-00032-t003:** Summary of the results for Waller-Duncan k ratio test for Zoometric characteristics in Creole chickens of Ecuador sorted by sex.

Variable	Male	Female
Head length (heal)	5.85 ^a^	5.42 ^b^
Crest length (crel)	3.50 ^a^	3.22 ^a^
Beak length (peal)	3.16 ^a^	2.98 ^b^
Neck length (necl)	14.19 ^a^	12.91 ^b^
Dorsal length (dorl)	23.12 ^a^	22.17 ^b^
Ventral length (venl)	24.49 ^a^	22.96 ^a^
Chest perimeter length (chpl)	33.45 ^a^	32.69 ^b^
Proximal humerus wing length (phwl)	10.21 ^a^	9.69 ^a^
Half wing radius ulna length (hwrul)	10.12 ^a^	9.82 ^a^
Distal phalanx wing length (dpwl)	8.43 ^a^	8.39 ^a^
Thigh length (femur) (thil)	12.25 ^a^	10.90 ^b^
Leg length (legl)	15.27 ^a^	14.35 ^b^
Leg circumference (legc)	9.23 ^a^	8.69 ^a^
Metatarsal tarsus length (metl)	9.08 ^a^	8.66 ^a^
Middle finger phalanx length (mfpl)	6.54 ^a^	6.10 ^b^

^a^ lower mean; ^b^ higher mean (*p* < 0.05). When there is not a significant difference between sexes superindex letters are the same (^a^).

**Table 4 animals-10-00032-t004:** Summary of results for Waller-Duncan test showing statistically significant differences in zoometric traits across provinces in Creole chickens of Ecuador.

Variable	Sample Size (N)	Bolívar	Chimborazo	Cotopaxi	Guayas	Morona Santiago	Tungurahua	Variation Coefficient (CV)	F	Pr > F
Head length (heal)	242	5.90 ^ab^	5.40 ^c^	4.38 ^d^	6.20 ^a^	5.68 ^bc^	5.60 ^bc^	15.00	16.48	<0.0001
Crest length (crel)	201	3.34 ^b^	3.06 ^b^	3.21 ^b^	3.88 ^a^	3.41 ^b^	3.12 ^b^	27.42	2.71	0.0150
Beak length (peal)	242	3.52 ^a^	2.70 ^bc^	2.83 ^b^	3.40 ^a^	3.47 ^a^	2.59 ^c^	16.06	27.24	<0.0001
Neck length (necl)	242	14.65 ^a^	12.51 ^b^	13.18 ^b^	13.39 ^b^	13.09 ^b^	12.75 ^b^	15.83	7.13	<0.0001
Dorsal length (dorl)	242	23.29 ^b^	22.44 ^b^	19.06 ^c^	23.12 ^b^	25.74 ^a^	19.91 ^c^	11.69	27.15	<0.0001
Ventral length (venl)	242	28.31 ^a^	24.34 ^b^	15.96 ^d^	26.67 ^a^	22.80 ^b^	20.38 ^c^	19.95	26.82	<0.0001
Chest perimeter length (chpl)	242	33.99 ^b^	31.08 ^c^	31.28 ^c^	37.00 ^a^	36.47 ^a^	30.16 ^c^	13.90	12.69	<0.0001
Proximal humerus wing length (phwl)	242	9.01 ^bc^	10.68 ^a^	10.06 ^ab^	9.31 ^bc^	9.61 ^abc^	8.81 ^c^	23.67	4.28	0.0004
Half wing radius ulna length (hwrul)	242	9.33 ^b^	10.76 ^a^	9.47 ^b^	9.87 ^ab^	9.83 ^ab^	8.88 ^b^	22.53	3.96	0.0008
Distal phalanx wing length (dpwl)	242	8.13 ^abc^	9.07 ^a^	8.37 ^ab^	7.38 ^bc^	9.07 ^a^	7.19 ^c^	24.59	5.25	<0.0001
Thigh length (femur) (thil)	242	10.86 ^b^	10.18 ^b^	12.41 ^a^	10.19 ^b^	10.94 ^b^	13.04 ^a^	15.81	17.38	<0.0001
Leg length (legl)	238	13.91 ^c^	11.87 ^d^	16.34 ^b^	14.06 ^c^	14.58 ^c^	19.77 ^a^	17.76	40.43	<0.0001
Leg circumference (legc)	232	8.72 ^c^	8.55 ^c^	6.48 ^d^	9.83 ^b^	11.09 ^a^	8.36 ^c^	23.39	15.06	<0.0001
Metatarsal tarsus length (metl)	242	8.78 ^b^	8.79 ^b^	5.12 ^c^	9.41 ^b^	10.62 ^a^	9.33 ^b^	17.08	43.85	<0.0001
Middle finger phalanx length (mfpl)	242	6.29 ^a^	6.33 ^a^	6.58 ^a^	5.63 ^b^	6.28 ^a^	5.61 ^b^	20.29	3.77	0.0013

^a,b,c^ Different letters in the superindex are indicative of the existence of significant differences among provinces (*p* < 0.05). If the same letter is present in different provinces then, no significant difference is found.

**Table 5 animals-10-00032-t005:** Multivariate statistics and F approximations for testing the significance of canonical correlations between zoometric variables and province classification variable.

Statistic	Value	F value	dfn	dfd	Pr > F
Pillai’s Trace	2.6973	17.7378	70	1060	<0.0001
Hotelling-Lawley Trace	9.9878	29.4736	70	729	<0.0001
Roy’s Greatest root	5.4219	82.1035	14	212	<0.0001
Wilk’s Lambda Ratio	0.0100	23.1461	70	994	<0.0001

dfn: degrees of freedom numerator; dfd: degrees of freedom denominator.

**Table 6 animals-10-00032-t006:** Multicollinearity analysis of biometric variables.

Statistic	Tolerance (1−R^2^)	VIF
Head length (heal)	0.5323	1.8788
Crest length (crel)	0.7304	1.3690
Crest width (crew)	0.7034	1.4217
Beak length (peal)	0.5330	1.8763
Neck length (necl)	0.6543	1.5282
Dorsal length (dorl)	0.3612	2.7685
Ventral length (venl)	0.4821	2.0741
Chest perimeter length (chpl)	0.5923	1.6883
Distal phalanx wing length (dpwl)	0.7053	1.4177
Thigh length (femur) (thil)	0.4929	2.0289
Leg length (legl)	0.4069	2.4579
Leg circumference (legc)	0.5591	1.7884
Metatarsal tarsus length (metl)	0.4458	2.2429
Middle finger phalanx length (mfpl)	0.6869	1.4559

Interpretation thumb rule: VIF = 1 (Not correlated); 1 < VIF < 5 (Moderately correlated); VIF ≥ 5 (Highly correlated).

**Table 7 animals-10-00032-t007:** Canonical variate pairs (discriminant functions) found in canonical discriminant analysis for zoometric variables.

Canonical Variate (Discriminating Functions)	Canonical Correlation	Squared Canonical Correlation	Eigenvalue	R^2^ (Explained Variance)
F1	0.919	0.844	5.422	0.543
F2	0.852	0.726	2.644	0.265
F3	0.696	0.485	0.941	0.094
F4	0.627	0.394	0.649	0.065
F5	0.499	0.249	0.331	0.033

An efficient model will report a vale of >0.4 for squared canonical correlations which translates into around 9% of explained variance among groups, provinces in our case.

**Table 8 animals-10-00032-t008:** Results for the tests of equality of group means to test for differences across provinces once redundant variables have been removed.

Variable	Wilks’ Lambda	F	df1	df2	*p*-Value	Rank
Head length (heal)	0.6938	19.5039	5	221	<0.0001	7
Crest length (crel)	0.9427	2.6888	5	221	0.0221	14
Crest width (crew)	0.7700	13.2047	5	221	<0.0001	8
Beak length (peal)	0.6377	25.1148	5	221	<0.0001	5
Neck length (necl)	0.8999	4.9189	5	221	0.0003	13
Dorsal length (dorl)	0.6442	24.4107	5	221	<0.0001	6
Ventral length (venl)	0.5598	34.7614	5	221	<0.0001	3
Chest perimeter length (chpl)	0.8010	10.9814	5	221	<0.0001	10
Distal phalanx wing length (dpwl)	0.8838	5.8128	5	221	<0.0001	11
Thigh length (femur) (thil)	0.7771	12.6790	5	221	<0.0001	9
Leg length (legl)	0.4965	44.8189	5	221	<0.0001	2
Leg circumference (legc)	0.5935	30.2791	5	221	<0.0001	4
Metatarsal tarsus length (metl)	0.4138	62.6212	5	221	<0.0001	1
Middle finger phalanx length (mfpl)	0.9094	4.4048	5	221	0.0008	12

F: Fisher-Snedecor approximation statistic; df1: numerator degrees of freedom; df2: denominator degrees of freedom, Rank denotes the importance of the discriminating power of a certain variable. As a rule of thumb, the closer Wilks’ lambda is to 0, the more the variable contributes to the discriminant function, hence placed at higher positions in the rank.

**Table 9 animals-10-00032-t009:** Standardized coefficients for zoometric variables.

Items	F1	F2	F3	F4	F5
Intercept (constant)	−3.6106	−3.6465	−2.0551	−2.3157	−2.9908
Head length (heal)	0.0923	0.0077	−0.0115	0.7568	−0.1707
Crest length (crel)	−0.3936	−0.1515	0.0834	−0.1946	−0.4764
Crest width (crew)	0.2127	0.1287	0.4228	−0.1815	0.1555
Beak length (picl)	0.1746	0.3402	0.4596	0.399	0.5273
Neck length (necl)	−0.3508	0.1704	0.3078	0.1698	−0.3943
Dorsal length (dorl)	−0.0489	0.2713	0.1360	−0.3311	−0.0388
Ventral length (venl)	0.0703	0.4387	−0.8659	0.3756	0.3443
Chest perimeter length (chpl)	−0.0793	0.0603	0.6318	0.0991	−0.3057
Distal phalanx wing length (dpwl)	0.0745	0.4506	−0.4368	−0.3062	0.2837
Thigh length (femur) (thil)	−0.1489	−0.6413	−0.1997	−0.174	0.5706
Leg length (legl)	0.2643	−0.6800	−0.083	0.0571	0.4576
Leg circumference (legc)	0.5107	0.2675	0.3097	−0.6428	0.0709
Metatarsal tarsus length (metl)	1.2468	-0.1421	−0.2844	−0.0454	−0.3439
Middle finger phalanx length (mfpl)	−1.0371	0.3027	0.007	−0.014	−0.0011

Linear combination for a discriminant function (Z) could be described by F1 (Z) = µ1 Y1 + µ2 Y2 + ... + µi Yi, where µi is the canonical coefficient, and Yi are independent variables measured. F1, F2, F3, F4 and F5: 1st, 2nd, 3rd, 4th and 5th discriminant functions.

**Table 10 animals-10-00032-t010:** Discriminant loadings or structure correlation for zoometrical variables across provinces.

Items	F1	Rank	VPVF	F2	Rank	VPVF	F3	Rank	VPVF	F4	Rank	VPVF	F5	Rank	VPVF	PCV
Head length (heal)	0.465	4	0.117	−0.025	11	0.000	0.221	7	0.005	0.503	1	0.016	0.009	11	0.028	0.139
Crest length (crel)	0.077	10	0.003	−0.184	12	0.009	0.223	6	0.005	0.031	8	0.000	−0.114	13	0.000	0.017
Crest width (crew)	0.304	7	0.050	0.290	5	0.022	0.282	5	0.008	−0.323	14	0.007	0.213	6	0.002	0.088
Beak length (picl)	0.233	9	0.029	0.342	3	0.031	0.463	1	0.020	0.457	2	0.014	0.429	2	0.006	0.100
Neck length (necl)	0.005	11	0.000	0.089	9	0.002	0.336	3	0.011	0.287	3	0.005	0.171	8	0.001	0.019
Dorsal length (dorl)	0.463	5	0.117	0.450	2	0.054	0.175	9	0.003	-0.102	10	0.001	0.188	7	0.001	0.175
Ventral length (venl)	0.325	6	0.057	0.609	1	0.098	−0.263	14	0.007	0.241	4	0.004	0.323	4	0.003	0.169
Chest perimeter length (chpl)	0.264	8	0.038	0.219	8	0.013	0.459	2	0.020	0.079	7	0.000	−0.057	12	0.000	0.071
Distal phalanx wing length (dpwl)	−0.018	12	0.000	0.303	4	0.024	−0.167	13	0.003	−0.294	12	0.006	0.087	9	0.000	0.033
Thigh length (femur) (thil)	−0.087	13	0.004	−0.416	13	0.046	0.194	8	0.004	−0.151	11	0.002	0.506	1	0.008	0.064
Leg length (legl)	0.522	3	0.148	−0.557	14	0.082	0.083	10	0.001	0.203	5	0.003	0.341	3	0.004	0.237
Leg circumference (legc)	0.566	2	0.174	0.263	6	0.018	0.323	4	0.010	−0.212	13	0.003	0.265	5	0.002	0.208
Metatarsal tarsus length (metl)	0.822	1	0.367	0.006	10	0.000	−0.124	11	0.001	0.040	6	0.000	−0.171	14	0.001	0.370
Middle finger phalanx length (mfpl)	−0.214	14	0.025	0.245	7	0.016	−0.127	12	0.002	−0.004	9	0.000	0.043	10	0.620	0.042

VPVF = Discriminant power of a certain variable within the function. PCV = Composite discriminant power of a certain variable. The composite power of each variable results after summing each of its particular effects on each discriminant function.

**Table 11 animals-10-00032-t011:** Distance of Mahalanobis between locations (above the diagonal) and F statistics (numerator degrees of freedom (dfn) = 6, denominator degrees of freedom (dfd) = 214) for the square distances between locations (below the diagonal).

Items	Bolívar	Chimborazo	Cotopaxi	Guayas	Morona Santiago	Tungurahua
Bolívar	0	11.4702	740.9480	43.5906	46.9021	60.1553
Chimborazo	33.4763	0	579.1246	53.3778	65.5743	72.6499
Cotopaxi	73.1381	53.4517	0	322.1637	135.4148	80.8406
Guayas	25.0129	9.1509	989.6791	0	17.9435	42.6445
Morona Santiago	35.1446	29.6095	1176.8949	48.2348	0	324.8541
Tungurahua	68.4382	51.4504	1984.7833	151.4674	110.0090	0

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
