# Peer review of "Sexual Dimorphism and Breed Characterization of Creole Hens through Biometric Canonical Discriminant Analysis across Ecuadorian Agroecological Areas"

_animals, 2019, doi:10.3390/ani10010032_

Round 1
Reviewer 1 Report
This investigation reports the biometrical characterization of Creole hens across different Ecuadorian ecological areas. The study is certainly very interesting, but it suffers some drawback in the present form.
According to the instructions of the Journal, Introduction should briefly place the study in a broad context, so the need of conservation of local genetic resources may be synthesized.
In Materials and Methods, the well-established methods should be briefly described and appropriately cited. This applies to Waller-Duncan test (lines 185-194) and, especially, to Canonical Discriminant Analysis (lines 207-346, too long and unfocused).
Results should be divided into subsections. If the results are reported in tables, it is not necessary to cite so many values in the main text (lines 348-409); it is very difficult to follow. In addition, Mahalanobis’ distance analysis could be cited and not described (lines 535-544).
In Discussion, it is difficult to follow interpretation of the results in perspective of previous studies, so this section should be synthesized and rearranged in some way (lines 571-638). Canonical Discriminant Analysis suggests the existence of four large blocs. Even if published very recently, the paper “Deciphering the Patterns of Genetic Admixture and Diversity in the Ecuadorian Creole Chicken” of the same Authors should be cited for an effectual comparison of the agroecological structure (genetic and environmental) with molecular information.
Some minor comments:
Line 83: replace “a.C.” with “A.D.” or, simply, delete it
Lines 172-174: the variables listed in Table 2 are 16.
Lines 472-473: it is the footer of Table 9
See references 31, 33, 65, 83, an 84; they are incomplete or they need to be revised
Author Response
Comments and Suggestions for Authors
This investigation reports the biometrical characterization of Creole hens across different Ecuadorian ecological areas. The study is certainly very interesting, but it suffers some drawback in the present form.
According to the instructions of the Journal, Introduction should briefly place the study in a broad context, so the need of conservation of local genetic resources may be synthesized.
Response: introduction was synthesized following the reviewer suggestion and reduced to 924 words from 1145 words.
In Materials and Methods, the well-established methods should be briefly described and appropriately cited. This applies to Waller-Duncan test (lines 185-194) and, especially, to Canonical Discriminant Analysis (lines 207-346, too long and unfocused).
Response: Waller-Duncan Section was reduced from 9 lines to 5 and Canonical Discriminant Analysis Section was reduced from 139 to 93 lines (or 1900 to 1321 words) as suggested by reviewer.
Results should be divided into subsections. If the results are reported in tables, it is not necessary to cite so many values in the main text (lines 348-409); it is very difficult to follow. In addition, Mahalanobis’ distance analysis could be cited and not described (lines 535-544).
Response: Results Section was subdividied into subsections matching those in the Material and Methods Section. Results were summarised to avoid body text repetitions of table data. Mahalanobis distances information was reduced to citation and simple interpretation thumb rule.
In Discussion, it is difficult to follow interpretation of the results in perspective of previous studies, so this section should be synthesized and rearranged in some way (lines 571-638).
Response: Discussion Section reported by reviewer was rearranged and summarized. We shortened the content from 1017 words to 849.
Canonical Discriminant Analysis suggests the existence of four large blocs. Even if published very recently, the paper “Deciphering the Patterns of Genetic Admixture and Diversity in the Ecuadorian Creole Chicken” of the same Authors should be cited for an effectual comparison of the agroecological structure (genetic and environmental) with molecular information.
Response: A paragraph was included in regards the comparison of molecular information as suggested by the reviewer.
Some minor comments:
Line 83: replace “a.C.” with “A.D.” or, simply, delete it
Response: We followed the reviewer suggestion and deleted “a.C.”
Lines 172-174: the variables listed in Table 2 are 16.
Response: Corrected.
Lines 472-473: it is the footer of Table 9
Response: Corrected.
See references 31, 33, 65, 83, an 84; they are incomplete or they need to be revised
Response: References were checked and corrected on the whole.
Reviewer 2 Report
The Authors focus on Creole hens bred in Ecuador. They study 14 FAO zoometric measures in 281 animals evaluated across 3 climatic regions and 6 provinces. The Canonical Discriminant Analysis was used to provide insights on the possible clustering patterns in Ecuadorian hen populations.
General comments
The study is interesting, especially for the mathematical approach, but the manuscript as such, at its present state, have to be improved.
Specific comments
In my opinion “Materials and Methods” section have to be reduced. Please move all the very interesting mathematical methods explanations in the “Discussion” section.
Line 139: please modify “….281 birds (244 hens (84.84%) and 37 roosters (15.16%)….” in “281 birds, 244 hens (84.84%) and 37 roosters (15.16%)…”.
Line 265: please modify “2.4.3. Canonical coefficients and loading interpretation and spatial representation” in “2.4.4. Canonical coefficients and loading interpretation and spatial representation”.
Line 323: please modify “2.4.4. Discriminant function cross-validation” in “2.4.5. Discriminant function cross-validation”.
“Results” section: in my opinion, all the comments have to be moved to “Discussion” section.
Line 421-423: please clarify if the comments are referring to this paper or a previously one.
Line 447: the legend of Table 8 have to be integrated: please specify the significance of “F”, “df1”, “df2” and “Rank”.
Line 453-460: please clarify the meaning of these sentences.
Line 470: in Table 9, the variables F1, F2, F3, F4 and F5 are calculated as above, but what is their meaning? Please specify the meaning of the variables in the legend.
Line 490: please some columns have to be enlarged because are difficult to reading.
Line 496 and line 504: it is possible to improve the style of the figure?
Line 532: please replace “Table 12” with “Table 12”
Author Response
Comments and Suggestions for Authors
The Authors focus on Creole hens bred in Ecuador. They study 14 FAO zoometric measures in 281 animals evaluated across 3 climatic regions and 6 provinces. The Canonical Discriminant Analysis was used to provide insights on the possible clustering patterns in Ecuadorian hen populations.
General comments
The study is interesting, especially for the mathematical approach, but the manuscript as such, at its present state, have to be improved.
Response: We thank for the reviewer comments as they improve the quality of the manuscript. The changes and suggestions outlined by the reviewer were approached as follows:
Specific comments
In my opinion “Materials and Methods” section have to be reduced. Please move all the very interesting mathematical methods explanations in the “Discussion” section.
Response: Material and Methods section was reduced from 2257 to 1537 words following reviewer’s suggestion.
Line 139: please modify “….281 birds (244 hens (84.84%) and 37 roosters (15.16%)….” in “281 birds, 244 hens (84.84%) and 37 roosters (15.16%)…”.
Response: Modified.
Line 265: please modify “2.4.3. Canonical coefficients and loading interpretation and spatial representation” in “2.4.4. Canonical coefficients and loading interpretation and spatial representation”.
Response: Subsections were completely renamed and renumber according to the suggestion of both reviewers.
Line 323: please modify “2.4.4. Discriminant function cross-validation” in “2.4.5. Discriminant function cross-validation”.
Response: Subsections were completely renamed and renumber according to the suggestion of both reviewers.
“Results” section: in my opinion, all the comments have to be moved to “Discussion” section.
Response: We followed reviewer’s comment and reduced the “Results” Section.
Line 421-423: please clarify if the comments are referring to this paper or a previously one.
Response: We clarified it.
Line 447: the legend of Table 8 have to be integrated: please specify the significance of “F”, “df1”, “df2” and “Rank”.
Response: Concepts were clarified.
Line 453-460: please clarify the meaning of these sentences.
Response: We clarified the meaning of the sentences as suggested by the reviewer.
Line 470: in Table 9, the variables F1, F2, F3, F4 and F5 are calculated as above, but what is their meaning? Please specify the meaning of the variables in the legend.
Response: Meaning added..
Line 490: please some columns have to be enlarged because are difficult to reading.
Response: We enlarged the columns to make text fit and enhance readability.
Line 496 and line 504: it is possible to improve the style of the figure?
Response: This is the best way that we found to represent it as the other possibilities were confusing.
Line 532: please replace “Table 12” with “Table 11”
Response: replaced.